# Neuroendocrine Effects of Carnitines on Reproductive Impairments

**DOI:** 10.3390/ijms221910781

**Published:** 2021-10-05

**Authors:** Tabatha Petrillo, Christian Battipaglia, Mohamed Ashraf Virmani, Andrea R. Genazzani, Alessandro D. Genazzani

**Affiliations:** 1Gynecological Endocrinology Center, Department of Obstetrics and Gynecology, University of Modena and Reggio Emilia, 41121 Modena, Italy; tabathap@libero.it (T.P.); christianbattipaglia@gmail.com (C.B.); 2Alfa Sigma, 3528 BG Utrecht, The Netherlands; Ashraf.Virmani@alfasigma.com; 3Department of Obstetrics and Gynecology, University of Pisa, 56126 Pisa, Italy; argenazzani@gmail.com

**Keywords:** functional hypothalamic amenorrhea, FHA, stress, metabolism, PCOS, carnitines

## Abstract

Carnitines are quaternary amines involved in various cellular processes such as fatty acid uptake, β-oxidation and glucose metabolism regulation. Due to their neurotrophic activities, their integrative use has been studied in several different physio-pathological conditions such as anorexia nervosa, chronic fatigue, vascular diseases, Alzheimer’s disease and male infertility. Being metabolically active, carnitines have also been proposed to treat reproductive impairment such as functional hypothalamic amenorrhea (FHA) and polycystic ovary syndrome (PCOS) since they improve both hormonal and metabolic parameters modulating the neuroendocrine impairments of FHA. Moreover, they are capable of improving the lipid profile and the insulin sensitivity in patients with PCOS.

## 1. What Are Carnitines?

Carnitines are quaternary amines present in almost all animal species introduced with food or synthesized at the cellular level. They can be considered as a vitamin-like substance [1,2] and there are two enantiomeric forms of carnitine: L-carnitine, which has an important role in cellular energy production, and D-carnitine, which is physiologically inactive and toxic. In addition to these carnitines there are various carnitines esters (acylcarnitines), like acetyl-L-carnitine, the prominent short-chain ester, or propionyl L-carnitine [3]. Since L-carnitine is produced mainly in the liver, kidneys and brain from the essential amino acids lysine and methionine [4], other tissues depend on carnitine uptake from blood. In fact, in subjects with impaired liver or kidney function, defects in L-carnitine synthesis can occur and consequently low carnitine concentrations can be found [5].

The primary source of L-carnitine is red meat, such as beef or lamb, but lower concentrations can be found in fish, pork, poultry and dairy products. Plant-origin products are not noteworthy sources of carnitines, except for avocado and asparagus and, because of that, strict vegetarian diets may lead to carnitine deficiency [6].

Carnitine is mainly absorbed by passive diffusion in the gastrointestinal tract through different membranes transporters called Organic Cation Transporters (OCTNs) [2]. It is not metabolized and it is freely filtered at the renal glomerulus and it is almost fully reabsorbed by the renal tubule. It is eliminated from the body via renal excretion in the form of L-carnitine, acetyl-L-carnitine (ALC) and other acylcarnitine esters [7]. The carnitine pool is primarily located within skeletal and myocardial muscle, though deposits are also at the liver and kidneys level.

## 2. Carnitine Functions

L-carnitine has a critical role in fatty acid uptake, in β-oxidation and in glucose metabolism regulation. Fatty acid metabolism is an essential source of energy in humans, especially in heart tissue. The main sources of fatty acids for the heart are free fatty acids (FFA), fatty esters transported in chylomicrons and very low-density lipoproteins (VLDL). As the fatty acids cross the sarcolemmal membrane, they are transformed into long-chain acyl-CoA by acyl-CoA synthetase. The acyl molecules are transferred in the mitochondria by a complex of enzymes involving carnitine palmitoyltranferase 1 (CPT1), carnitine acyltranslocase and carnitine palmitoyltranferase 2 (CPT2). L-carnitine is a cofactor of CPT 1, acting as an acceptor of fatty acyl groups from acyl-CoA to long-chain fatty acyl carnitine. The latter is transported into the mitochondrial matrix by carnitine acyltranslocase where CPT2 catalyzes the reverse reaction regenerating long-chain acyl-CoA. Finally, acyl-CoA undergoes mitochondrial β-oxidation producing acetyl-CoA [8].

The other main source of mitochondrial acetyl-CoA is oxidation of carbohydrates (glucose and lactate). As soon as glucose enters the cell membrane, it is converted to glucose-6-phosphate in the cytosol, which is later, through glycolysis, transformed into two molecules of pyruvate. The pyruvate enters the mitochondrial membrane and forms acetyl-CoA through pyruvate dehydrogenase (PDH) complex activity.

The acetyl-CoA obtained by pyruvate dehydrogenase (PDH) or fatty acid β-oxidation enters the tricarboxylic acid (TCA) cycle, producing energy in ATP form. The PDH complex is the key rate-limiting step in carbohydrate oxidation and its activity is regulated by the concentrations of substrates (pyruvate and CoA) and products (acetyl-CoA). When the ratio of acetyl CoA/CoA increases, PDH is inhibited leading to the reduction of the glucose oxidation rate. The acetyl CoA/CoA ratio is determined by the rate of removal of intramitochondrial acetyl-CoA by the TCA cycle and the rate of fatty acid β-oxidation [9].

L-carnitine can also transport acetyl groups from the mitochondrial matrix to the cytoplasm. In the mitochondria, carnitine acetyltransferase catalyzes the transfer of acetyl groups from acetyl-CoA to carnitine forming acetyl carnitine, which is then transferred into the cytoplasm where the acetyl groups are transferred back onto CoA [8] (Figure 1).

In conclusion, carnitine modulates the intramitochondrial acetyl CoA/CoA ratio as it regulates the supply of acetyl-CoA from both the PDH complex and from β-oxidation of fatty acids. Therefore, carnitine plays a substantial role in both carbohydrate and lipid metabolism.

Additionally, L-carnitine is important in maintaining cell membrane stability through its involvement in acetylation of membrane phospholipids and amphiphilic actions [10,11]. It also prevents DNA damage induced by the detrimental actions of free radicals [12]. Moreover, carnitine supplementation has been found to decrease pro-inflammatory cytokines such as interferon-γ, tumor necrosis factor-α (TNF-α), and interleukins-2 (IL-2) and -6 (IL-6), with an anti-inflammatory effect [13].

Carnitines also promote cellular proliferation and decrease apoptosis due to their stimulating effect on mitochondria and by inhibiting TNFα and other anti-proliferative agents [14,15] (Figure 2).

## 3. Clinical Effects of Integration with Carnitines

Exogenous carnitine supplementation has been reported to improve different pathological conditions such as anorexia, male infertility, chronic fatigue, vascular diseases, Alzheimer’s disease, hypothalamic amenorrhea and polycystic ovary syndrome.

Reactive oxygen species (ROS) are highly reactive chemical molecules formed as a natural byproduct of the normal aerobic metabolism of oxygen [16]. It is well known that ROS are produced by biochemical reactions that occur during the processes of respiration in mitochondria [17]. Many impairments and much damage can be triggered by excessive ROS production, such as infertility, especially in males [18], and, later, ageing. When ROS levels exceed the buffering capacity of the cells, the sum of their deleterious reactions trigger and constitute the ageing process, particularly in the brain as it can compromise synaptic plasticity and cognitive functions [19].

Acetyl-L-carnitine (ALC) has the ability to cross blood–brain barriers and has a powerful antioxidant effect, preventing brain cell deterioration. It has been proven that ALC improves the energy metabolism of neurons and facilitates the release and synthesis of acetylcholine and maintains cellular membrane stability. Therefore, ALC can be used to enhance both short and long-term memory and develop the attention span. ALC supplementation slows the progression of Alzheimer’s disease and significantly ameliorates response times in patients with cerebrovascular insufficiency [20].

While ALC appears to be the preferable choice to use for Alzheimer’s disease and other brain disorders, propionyl-L-carnitine (PLC) seems to be the most effective for chest pain, ischemic heart disease, congestive heart failure, hypertrophic heart disease and peripheral vascular disease [2].

Absolute or relative carnitine deficiency is associated with development of chronic congestive heart failure, hypertrophic cardiomyopathy and acute myocardial ischemia. PLC has shown efficacy in treating these disorders, increasing glucose oxidation and/or providing propionyl groups as a source of carbon for the TCA cycle [8].

L-carnitine administration has been shown to improve male infertility since it serves as an intra-mitochondrial vehicle for the acyl group, which in the form of acyl-CoA acts as a substrate for the oxidation process, producing energy for sperm motility and respiration [21]. The carnitine/acetylcarnitine complex acts as an antioxidant agent and exerts a repairing effect through the removal of elevated intracellular toxic acetyl-coenzyme A and/or the replacement of fatty acids in membrane phospholipids, minimizing pathological disorders of sperm [22].

Oxidative stress is also involved in female reproduction as it affects the cumulus-oocyte complex (COC) and decreases embryo development and blastocyst fragmentation. Free L-carnitine and ALC serve three important functions in oocytes: first, they increase energy production by transferring palmitate into mitochondria and maintaining the acetyl-CoA/CoA ratio; secondly, they reduce oxidative stress and lipotoxicity by scavenging free radicals and removing excess palmitate from the endoplasmic reticulum (ER) and finally they promote oocyte growth and maturation by decreasing the rate of apoptosis [12,15,23,24] (Figure 3). Such positive effects of carnitines have been recently demonstrated in an animal model of PCOS [25].

As we previously addressed, ALC facilitates the release and synthesis of acetylcholine and modulates the gamma-amino butyric acid (GABA) system. Therefore, it can affect the neuronal activity of the hypothalamus–pituitary–gonadal (HPG) axis, enhancing female reproduction [27]. Carnitine metabolism regulates hypothalamic fatty acid sensing through the actions of CPT 1 and glucose sensing modulating levels of acetyl-CoA. Therefore, it represents a key molecular target that can simultaneously integrate nutrient and hormonal information and is critical to maintain energy homeostasis [28]. For these reasons, carnitine supplementation has been successfully used to treat functional hypothalamic amenorrhea and polycystic ovary syndrome.

Functional hypothalamic amenorrhea (FHA) is one of the most common causes of secondary amenorrhea. This reproductive impairment occurs as a defensive mechanism triggered to react to stressors such as food restriction, weight loss, psychological stress or excessive physical exercise [29]. The stressors induce a blockade of spontaneous release of GnRH, thus affecting pituitary secretion of gonadotropins and then of the ovarian cycle [30].

Amenorrhea in general, and especially in patients with FHA, induces a hypo estrogenic condition that negatively affects most of the estrogen-sensitive organs. For this reason, it has to be resolved within a reasonable span of time. The occurrence of osteopenia, as well as mood and/or behavior disorders and vaginal atrophy are common clinical signs reported by these patients. Removal of stressors and/or hormonal treatment are the logical therapeutic strategies that can be applied [31].

Various putative treatments have been suggested to restore the activity of the reproductive axis, ranging from estroprogestins, estradiol and estriol to neuroactive compounds such as pivagabine and naltrexone, a competitive antagonist of the μ-opioid receptor [31,32,33,34].

Patients affected by FHA often show higher levels of prolactin (PRL) and/or cortisol, while LH plasma levels are below normal threshold, showing an abnormal pulsatile secretory pattern [35].

Experimental studies also indicate that the increased release from hypothalamic neurons of corticotrophin-releasing hormone (CRH) may play a role in stress-related reproductive dysfunction in both sexes. In fact CRH is able to reduce GnRH release from the medio basal hypothalamus in vitro and the intraventricular infusion of CRH reduces GnRH and LH release in rats [36].

Nappi et al. [37] reported that women with hypothalamic amenorrhea have a cortisol secretory pattern higher than normal women. They have an impaired opiatergic-serotoninergic- induced release of cortisol and a blunted cortisol response to the CRH test due to a stress-induced higher hypothalamic CRH tone. Since opioid peptides blunt GnRH release from the GnRH-secreting neurons of the hypothalamus, the use of the naloxone (an antagonist of opioid receptors) test can unmask such a blockade, if really present. The lack of a cortisol response to naloxone infusion may be correlated with an increased activation of central opioid pathways. In fact, CRH release due to stress, stimulates hypothalamic β-endorphin neurons that, in turn, inhibit GnRH release. The same lack of cortisol response has been reported after the administration of fenfluramine, which is a serotoninergic agonist. In normal subjects, fenfluramine determines a significant increase of plasma ACTH and cortisol levels. These findings support the hypothesis that stress induces HPA axis hyperactivity. The increased CRH secretion is associated with an increased activity of opioids and serotoninergic pathways, which have an inhibitory action on GnRH neurons [37].

In addition, Genazzani et al. [38] showed that the administration of the long-acting blocker of opioidergic receptors, naltrexone, for 9 months increased mean estradiol and LH plasma levels in all patients with hypothalamic amenorrhea and BMI < 20, and restored menstrual bleeding in 24 of 30 patients (80%). This data supports the efficacy of naltrexone in the treatment of hypothalamic amenorrhea associated with weight loss and enforces the role of the opioidergic inhibitory tone on GnRH secretion [38].

It has to be pointed out that insulin and kisspeptin are known to be closely involved in the modulation of the neuroendocrine mechanisms of the reproductive axis since their plasma concentrations are severely impaired in FHA and show specific correlations with BMI. In addition patients with FHA and/or anorexia nervosa/bulimia show amylase levels higher than the normal population due to an excessive secretion of the salivary-type amylase [39].

On the basis of several studies on experimental animals showing that ALC was able to blunt the negative effects of stress-induced β-endorphin release on hypothalamic neuroendocrine functions [40,41], various trials evaluated the role of carnitines as a therapeutic agent in FHA. Genazzani et al. [30] studied 24 patients with FHA and BMI in the normal range who were tested for hormonal and metabolic parameters and underwent the naloxone test, before and after 16 weeks of ALC administration (1 g/day). Though this was not a randomized study, after the treatment interval, all patients showed a significant increase of LH and insulin plasma levels, but only the patients showing LH ≤ 3 mIU/mL (hypo-LH) before the ALC administration showed a significant rise of 17-OHP, basal LH and LH response to the naloxone test. These probable hypo-LH patients had a more heavily stress-induced impairment of the GnRH–LH axis. These results support the hypothesis that in humans ALC is effective in modulating LH response to GnRH, facilitating LH synthesis, storage and secretion from the pituitary [30,42], since previous reports demonstrated that ALC was able to blunt the negative effect of β-endorphin circadian rhythms in rats exposed to different stressors [40] and was able to increase gonadotropin content in the pituitary and both gonadotropin plasma levels and gonadotropin response to GnRH stimulation test in patients with FHA [42]. It was supposed that ALC acts through the opioidergic pathway modulating protein/hormone functions by acetylating -OH groups in amino acids like serine, threonine or tyrosine, thereby improving their functions [30].

Recently, two studies evaluated amenorrhoeic patients treated with a combination of L-carnitine (500 mg/day), ALC (250 mg/day), L-arginine (500 mg/die), N-acetyl cysteine (50 mg/die), and Vitamin E and C for 12 weeks [29,43]. The only significant changes observed were the increase in insulin and the decrease in amylase plasma levels. When patients were evaluated according to basal LH plasma levels, specific changes were reported in patients with LH ≤ 3 mIU/mL, after the integrative treatment, a significant increase of fT3, insulin and LH and a significant decrease of cortisol and amylase were observed [29,43].

Indeed according to recent studies, carnitines are mediators of the actions of metabolically active hormones such as ghrelin, leptin and insulin on NPY and POMC secreting neurons [28,44]. It can thereby be supposed that the combination of L-carnitine plus ALC acted positively on the neuropeptide Y (NPY) and pro-opiomelanocortin (POMC) secreting neurons that are sensitive to nutrients and are connected with the hypothalamic centers (Figure 4). The positive action on amylase and fT3 can be ascribed to the metabolic changes induced by the administration of carnitines [39].

This study disclosed the important role played by N-acetyl cysteine and L-arginine that, together with carnitines, interfere with ROS overproduction caused by excessive dieting and/or excessive physical exercise, improving the recovery of the endogenous anti-oxidant system such as glutathione [45].

## 4. Polycystic Ovary Syndrome

Polycystic ovary syndrome (PCOS) is a reproductive disorder that occurs in about 4–25% of women of reproductive age [46]. PCOS is clinically evident when at least two out of three of the Rotterdam criteria are present, that is: (1) oligo and/or anovulation (defined as delayed menses > 35 days); (2) clinical and/or biochemical signs of hyperandrogenism; (3) polycystic ovaries at ultrasound (12 or more follicles in each ovary with a diameter of 2–9 mm, and/or increased ovarian volume > 10 mL) [47]. The dismetabolic state of insulin resistance (IR) and its correlate compensatory hyperinsulinemia are frequently occurring features of the syndrome [45].

Moreover, it has been shown that there is a greater risk of mental health disorders such as depression and anxiety in women with PCOS. Abnormal psycho/behavioral aspects in these women might be related to abnormal synthesis of neurosteroids and impaired adrenal functions [48]. The standard approaches to treat women with PCOS vary from oral contraceptives with/without anti-androgenic drugs to insulin sensitizers such as metformin or complementary supplements such as inositols, alpha-lipoic acid, L-arginine (L-ARG), N-acetylcysteine (NAC) and carnitines [49,50]. PCOS is known to be associated with oxidative stress that increases the production of free radicals and reactive oxygen species (ROS) with decreased total antioxidant levels. A significant reduction in oxidative stress was observed following supplementation with 2 g/day of L-carnitine for 3 months in patients with type 2 diabetes mellitus [51]. Such positive effects of carnitines have been recently demonstrated in the PCOS rat model where the administration of L-carnitine (LC) and acetyl-L-carnitine (ALC) and propionyl-L-carnitine (PLC) greatly improved antioxidant molecular pathways in the ovarian microenvironment [25].

In addition, Jamilian et al. showed that carnitine supplementation of 250 mg/day for 12 weeks in patients with PCOS had favorable effects on parameters of mental health, evaluated by General Health Questionnaire (GHQ) and Depression Anxiety Stress Scales (DASS) scores, and on biomarkers of oxidative stress such as increased plasma total antioxidant capacity (TAC) and decreased malondialdehyde (MDA) [52]. Carnitine intake may decrease oxidative stress, stabilizing various cell membranes, increasing concentration of antioxidant enzymes, inhibiting microsomal peroxidation and preventing fatty acid membrane peroxidation [52].

Fenkci’s group demonstrated that non-obese women with PCOS have significantly lower total serum L-carnitine levels and higher levels of dehydroepiandrosterone (DHEA), testosterone (T), LH, low-density lipoproteins (LDL) and fasting insulin compared to healthy women [53]. In fact, this author reported that L-carnitine plasma levels were 50% lower and androgens (i.e., DHEAS and T) were double that of healthy controls. Reduced circulating and tissue carnitine levels, probably due to impaired mitochondrial function, have been postulated to be involved in the pathogenesis of insulin resistance. Molfino et al. demonstrated that L-carnitine administration, associated with a hypocaloric feeding regimen, improves insulin resistance and may represent an adjunctive treatment for impaired fasting glucose (IFG) and Type 2 diabetes mellitus (DM-2) [54].

Though using a low daily dosage (250 mg per day for 12 weeks), Samimi et al. [55] found that L-carnitine supplementation lead to a significant reduction in body weight, BMI, waist and hip circumference, as well as improved glycemic control in women with PCOS. This study supports the idea that L-carnitine supplementation improves PCOS by decreasing blood glucose levels and reducing insulin resistance, due to the effects on beta oxidation of fatty acids and carbohydrate metabolism [55,56]. Insulin resistance seems to be related to mitochondrial dysfunction, which, at least in part, is due to reduced fatty acid oxidation [57]. Inefficient oxidative phosphorylation induces the increase of oxidative stress and especially triglyceride accumulation in skeletal muscles, which reduces insulin sensitivity [54,58]. The biochemical mechanisms responsible for lower fatty acid oxidation involve reduced carnitine palmitoyltransferase (CPT) activity [58] and exogenous carnitine supplementation may restore the deficit.

Recently, a group of 53 overweight/obese PCOS patients were treated with an integrative mix composed of ALC 250 mg, LC 500 mg, L-ARG 500mg and NAC 50 mg each day for 6 months [45]. In all patients, a significant improvement was seen in metabolic parameters, in particular for total cholesterol, HDL, triglycerides, plasma insulin levels and HOMA index. No changes were observed in the levels of reproductive hormones. Moreover, the insulin response to the glucose load, as well as the hepatic insulin extraction index (HIE), decreased significantly. This index reflects the dynamics of both insulin and C peptide molecules, which derive from the cleavage of the same molecule of proinsulin released by pancreatic cells. While C peptide reflects the β-cells secretory capacity and its hepatic extraction is negligible, insulin is mainly cleared by the liver. The ratio between insulin and C peptide is theoretically 1 but in vivo the ratio computed between insulin and C peptide concentrations reflects their clearance kinetics. This study showed that hyper-insulinemic patients, that is those that showed a response above 50 µU/mL of insulin plasma levels within 90 min of glucose load [45] at the oral glucose tolerance test (OGTT), had a reduced insulin hepatic extraction under baseline conditions. In other words, the integrative treatment with carnitines at a higher daily dosage than Samimi et al. [55], together with specific anti-oxidants such as NAC and LArg greatly improved hepatic function and insulin degradation together with increased peripheral insulin sensitivity and a reduced insulin secretion [45]. In other words, this mix has a great metabolic effect due to the combined action of carnitines with that of NAC and LArg, which improves the regeneration of glutathione. According to this recent study, LArg-induced nitric oxide (NO) synthesis together with thiol groups released by NAC improve nitroso-cysteine (CysNO) synthesis, thus improving insulin sensitivity and also hypertensive predisposition [45].

## 5. Conclusions

Carnitines, individually or in combination with other supplements, can be used in different clinical scenarios as they have an important anti-oxidant effect and they are involved in regulating lipid and glucose metabolism and stabilizing cell membranes. Their effects are complex and this is demonstrated by a large variety of studies in animal models as well as in humans (Table 1). Though up to now, specific treating dosages have not been defined, an average treating dosage seems to be 500 mg every day. Carnitines can be administered to treat different pathological conditions such as chronic fatigue, vascular diseases, and Alzheimer’s disease, but there is some evidence that they are effective on metabolic and/or neuroendocrine impairments of the female reproductive axis. In fact, in hypothalamic amenorrhea, carnitines ameliorate both hormonal and metabolic parameters, while in patients with polycystic ovary syndrome (PCOS), their use together with anti-oxidant compounds greatly improves the lipid profile reducing insulin-resistance and the risk of non-alcoholic fat liver disease (NAFLD).

## Figures and Tables

**Figure 1 ijms-22-10781-f001:**
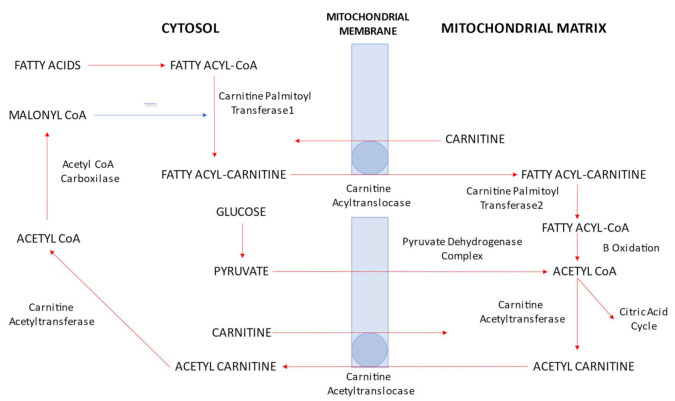
Role of carnitine in fatty acid uptake, β-oxidation and glucose metabolism regulation. CoA: coenzyme A.

**Figure 2 ijms-22-10781-f002:**
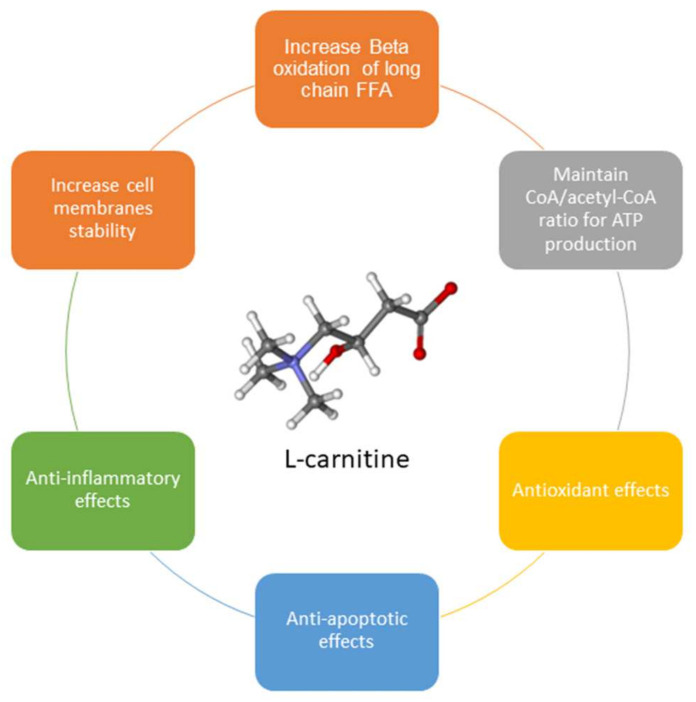
L-carnitine’s different functions.

**Figure 3 ijms-22-10781-f003:**
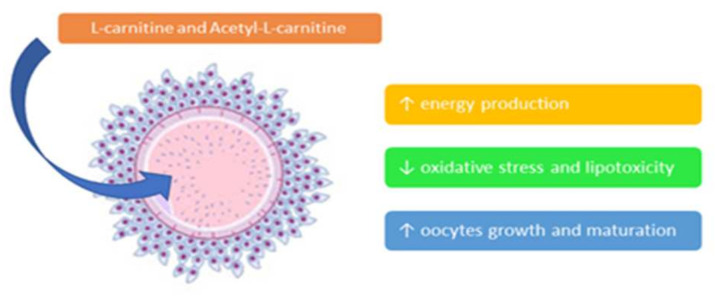
Role of L-carnitine and acetyl-L-carnitine on the cumulus-oocyte complex. Várnagy et al., in 2013, studied carnitine profiling of women undergoing IVF and suggested that L-carnitine metabolism is accelerated and the developmental competence of oocytes and early embryos can be optimized by giving supplemental L-carnitine [26].

**Figure 4 ijms-22-10781-f004:**
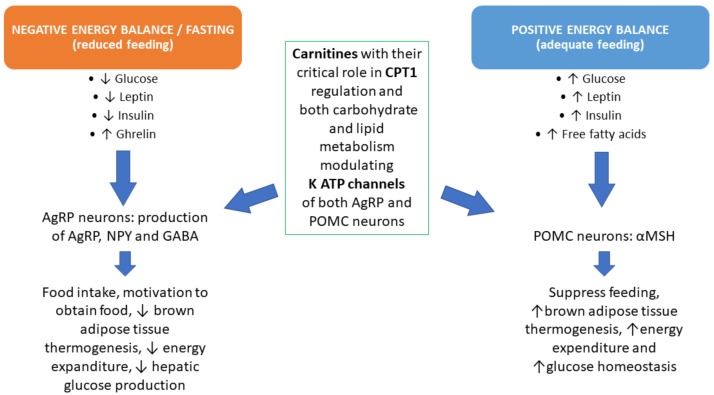
Feeding and related metabolic events are able to modulate hormones and neuropeptides. Normal feeding (**right**) or reduced feeding (**left**) act specifically through the activation of tissues and organs to compensate for any metabolic excess or deficiency. Carnitine integration permits specific modulations on most of these metabolic pathways through CPT1 regulation. **CPT1**: Carnitine Palmitoyl-Transferase 1; **AgRP**: Agouti-Related Protein; **NPY**: NeuroPeptide Y; **GABA**: Gamma Amino-Butyric Acid; **POMC**: Pro-Opio-Melano-Cortin; **MSH**: Melanocyte Stimulating Hormone.

**Table 1 ijms-22-10781-t001:** Most relevant studies that disclosed the role of carnitines on stress and on the reproductive axis.

	Reference	Dose and Duration	Study Design/Subjects	Outcomes
**Stress in animal model**	Bidzinska B et al. J Neuroendocrinol. 1993 [41]	ALC per os 10 mg/day/rat for 10 days at night.	Male Wistar rats were housed in hanging cages in groups of 6 in a light-controlled room (14 h light/ 10 h dark); temperature 20 °C. Food and water were available ad libitum.	ALC treatment was able to reverse the pituitary β-EP changes induced by stress.
**Infertility in animal model**	Krsmanović LZ et al. J Steroid Biochem Mol Biol. 1992 [27]	50 mg pro kg/die of ALC for 2 consecutive estrous cycles	Female Sprague-Dawley rats of 3 months of age	Improved hormonal secretions through HPG axis: increased GnRH, LH, Estradiol and progesterone levels
**Functional hypothalamic amenorrhea**	Genazzani AD et al. Acta Obstet Gynecol Scand. 1991 [42]	ALC (2 g/day, per os) for 6 months	Twenty patients with hypothalamic amenorrhea subdivided in two groups according to their LH plasma levels: hypogonadotropic with plasma LH less than 3 mIU/mL and normogonadotropic with plasma LH greater than 3 mIU/mL.	The hypogonadotropic subjects showed an increase in baseline plasma LH levels, an increase in LH pulse amplitude with no changes in LH pulse frequency, and an increased response of LH to the latter GnRH bolus during the GnRH test. Hypogonadotropic patients also showed a significant increase in both estradiol and PRL. No significant differences were observed in the hormonal parameters of normogonadotropic patients.
	Genazzani AD et al. J Endocrinol Invest. 2011 [30]	ALC (1 g/day, per os) for 16 weeks.	Twenty-four patients affected by stress-induced HA were divided into two groups according to LH plasma levels: hypo-LH (LH ≤ 3 mIU/mL; no. = 16), and normo-LH (LH > 3 mIU/mL; no. = 8).	Hypo-LH patients showed a significant increase in LH plasma levels and in LH pulse amplitude. No changes were observed in the normo-LH group. LH response to naloxone was restored under ALC therapy. Maximal LH response and area under the curve under naloxone were significantly increased.No changes were observed in the normo-LH patients.
	Genazzani AD et al. Gynecol Endocrinol. 2017 [43]	Combined integrative treatment for 12 weeks of ALC (250 mg/die) and L-carnitine (500 mg/die)	Twenty-seven patients with FHA.	Significant increase of LH plasma levels and the significant decrease of both cortisol and amylase plasma levels. The increased 17OHP/cortisol ratio, as index of the adrenal activity, demonstrated the reduced stress-induced adrenal activity.
	Genazzani AD et al. European Gynecology and Obstetrics. 2020 [29]	L-carnitine (500 mg) and acetyl-L-carnitine (250 mg) combined with LArg (500 mg), NAC (50 mg), and vitamins E and C as antioxidants, were administered daily for 12 weeks.	Twenty-nine patients with FHA.	LH and insulin increased, while amylase and cortisol decreased.
**Polycystic Ovary Syndrome**	Malaguarnera M et al. Am J Clin Nutr. 2009 [51]	The 2 groups received either 2 g L-carnitine once daily or placebo.	Eighty-one patients with diabetes were randomly assigned to 1 of 2 treatment groups for 3 months.	L-carnitine-treated patients showed significant improvements compared with the placebo group in the following markers: LDL levels, triglycerides, apolipoprotein A1 and apolipoprotein B-100 concentrations decreased.
	Molfino A. et al. J Parenter Enteral Nutr. 2010 [54]	Hypocaloric diet or the same dietetic regimen in addition to oral L-carnitine (2 g twice daily) supplementation for 10 days.	Sixteen Patients were randomly assigned to two groups	OGTT at 2 h improved in both groups. Only in the L-carnitine-supplemented group did plasma insulin levels and HOMA-IR significantly decrease when compared to baseline values.
	Samimi M et al. Clin Endocrinol (Oxf). 2016 [55]	250 mg carnitine supplements or placebo for 12 weeks.	Sixty overweight patients diagnosed with PCOS; randomized.	Taking carnitine supplements resulted in a significant reduction in weight, BMI, waist circumference and hip circumference compared with placebo. In addition, carnitine administration in women with PCOS led to a significant reduction in fasting plasma glucose, serum insulin levels, homoeostasis model of assessment-insulin resistance and dehydroepiandrosterone sulphate.
	Jamilian H et al. Gynecological Endocrinology. 2017 [52]	250 mg carnitine supplements or placebo for 12 weeks.	Sixty patients diagnosed with PCOS were randomized to take either carnitine supplements (n = 30) or placebo (n = 30).	Carnitine supplementation resulted in a significant improvement in Beck Depression Inventory total score, General Health Questionnaire scores and Depression Anxiety and Stress Scale scores.
	Genazzani AD et al. Gynecol Reprod Endocrinol Metab 2020 [45]	ALC (250 mg), L-carnitine (500 mg), L-arginine (500 mg) and N-acetyl cysteine (50 mg) were administered daily for 24 weeks	Fortyfive overweight/obese PCOS patients underwent daily integrative administration and were evaluated before and after 12 and 24 weeks of treatment.	After 12 and 24 weeks of treatment, all the subjects showed significant reduction of plasma insulin levels. Trygliceride, total cholesterol and HOMA index decreased while HDL increased significantly. On oral glucose tolerance testing, 39 out of the 45 PCOS patients showed a hyperinsulinemic response. This latter group showed the greatest significant reduction of all metabolic parameters and of the hepatic insulin extraction index (HIE). No changes were observed in normoinsulinemic PCOS patients (6 out of 45).

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
