# Peer review of "Neuroendocrine Effects of Carnitines on Reproductive Impairments"

_ijms, 2021, doi:10.3390/ijms221910781_

Round 1
Reviewer 1 Report
The article by Petrillo et al. is a review on the neuroendocrine effects of carnitines on some pathologies of the reproductive system such as hypothalamic amennorrhea and polycystic ovary syndrome. The authors after a brief exposure on carnitines, and what their mechanism of action is, explain what are the clinical effects of carnitine supplementation.
The article is clear enough but needs some clarification.
Page 5 line 128-135. The hypothesis about the role of carnitines in the oocyte needs to be reported by citing the original research papers instead of a review.
Recently, a paper regarding administration of different mixtures of acyl-L-carnitine in a PCOS mouse model has been published (doi 10.3390/antiox9090867). I strongly recommend to discuss it under the "PCOS” paragraph.
Please, provide sharper images.
English revision and correction of typing errors are recommended.
Please, replace “et all.” with “et al.” throughout the text.
Author Response
Reviewer # 1
The article by Petrillo et al. is a review on the neuroendocrine effects of carnitines on some pathologies of the reproductive system such as hypothalamic amennorrhea and polycystic ovary syndrome. The authors after a brief exposure on carnitines, and what their mechanism of action is, explain what are the clinical effects of carnitine supplementation.
The article is clear enough but needs some clarification.
- Page 5 line 128-135. The hypothesis about the role of carnitines in the oocyte needs to be reported by citing the original research papers instead of a review.
Recently, a paper regarding administration of different mixtures of acyl-L-carnitine in a PCOS mouse model has been published (doi 10.3390/antiox9090867). I strongly recommend to discuss it under the "PCOS” paragraph. - Please, provide sharper images.
- English revision and correction of typing errors are recommended.
- Please, replace “et all.” with “et al.” throughout the text.
Rebuttal
- The comment of the reviewer is correct but we cited the review to reduce the number of references cited and to permit to readers to have a precise reference to refer so that to have a clearer comprehension of this complex topic. We added the reference suggested since it appears to be the most appropriate for the context of the PCOS topic.
- Fig 1 has been redrawn
- English has been revised
- “et all.” has been replaced with “et al.” all along the manuscript.
Reviewer 2 Report
This is an interesting review of the effects of carnitines in patients with reproductive disorders such as polycystic ovary syndrome and functional hypothalamic amenorrhoea. Overall, nicely written review with the first figure being useful to summarise complex processes.
The review would benefit from a table that summarises the studies undertaken in each reproductive disorder, the sample size, intervention received, primary outcomes measured and the magnitude of change with confidence intervals. It would be helpful to have a graphic diagram of the proposed interaction between carnitines and ghrelin, leptin and insulin on NPY and POMC secreting neurons
Figure 3 is a little sparse.
The mechanism of action and the evidence for this was a little unclear for HA and PCOS (better for PCOS than HA). It would be good to include more details on the interventional studies and critically appraise them more fully, ie placebo controlled? Be specific as to what degree of improvement in what parameter was noted over what time frame ?
It would also be useful to discuss / summarise the dose and formulation that should be recommended based on the data available and whether this is important having reviewed the individual studies.
Occasional grammatical improvements can be made.
Line 100 add refs. For this paragraph, can more detail about what the studies actually were and what improvements were observed be provided as it currently reads like an infomercial ?
Line 165 and 189 why is Estradiol capitalized ?
Line 168 - is the high PRL reflective of stress in this context. Were cannulated prolactin conducted? Low E2 in HA can lead to lower PRL levels also.
Line 178 Is the blunted response to CRH indicative of high basal CRH tone?
Line 204 Please describe the utility of the naloxone test for non-expert readers. It is not a commonly performed test.
Line 212 Can the mechanism of action for how the ALC works in HA be described more clearly? It is a little difficult to follow.
Line 224 Was there data on return of cycles / ovulation rates? Any safety considerations in a fertility context?
Line 218 These are amino acids. Please clarify- Improving their function on what? some kind of receptor?
Line 228- What evidence is there for an action on POMC / AgRP neurons?
Line 239 Usually 20% is used as a generous upper limit for prevalence of PCOS. Please provide reference if 25% is to be put forwards.
line 269 add 'levels'
Line 277- this is helpful, but the quantity of improvement in which parameter and if placebo controlled would be helpful to add. Please critically appraise the studies included.
Likewise for the study in next paragraph.
Author Response
Reviewer # 2
This is an interesting review of the effects of carnitines in patients with reproductive disorders such as polycystic ovary syndrome and functional hypothalamic amenorrhoea. Overall, nicely written review with the first figure being useful to summarise complex processes.
- The review would benefit from a table that summarises the studies undertaken in each reproductive disorder, the sample size, intervention received, primary outcomes measured and the magnitude of change with confidence intervals. It would be helpful to have a graphic diagram of the proposed interaction between carnitines and ghrelin, leptin and insulin on NPY and POMC secreting neurons
- Figure 3 is a little sparse.
- The mechanism of action and the evidence for this was a little unclear for HA and PCOS (better for PCOS than HA). It would be good to include more details on the interventional studies and critically appraise them more fully, ie placebo controlled? Be specific as to what degree of improvement in what parameter was noted over what time frame ?
- It would also be useful to discuss / summarise the dose and formulation that should be recommended based on the data available and whether this is important having reviewed the individual studies.
- Occasional grammatical improvements can be made.
- Line 100 add refs. For this paragraph, can more detail about what the studies actually were and what improvements were observed be provided as it currently reads like an infomercial ?
- Line 165 and 189 why is Estradiol capitalized ?
- Line 168 - is the high PRL reflective of stress in this context. Were cannulated prolactin conducted? Low E2 in HA can lead to lower PRL levels also.
- Line 178 Is the blunted response to CRH indicative of high basal CRH tone?
- Line 204 Please describe the utility of the naloxone test for non-expert readers. It is not a commonly performed test.
- Line 212 Can the mechanism of action for how the ALC works in HA be described more clearly? It is a little difficult to follow.
- Line 224 Was there data on return of cycles / ovulation rates? Any safety considerations in a fertility context?
- Line 218 These are amino acids. Please clarify- Improving their function on what? some kind of receptor?
- Line 228- What evidence is there for an action on POMC / AgRP neurons?
- Line 239 Usually 20% is used as a generous upper limit for prevalence of PCOS. Please provide reference if 25% is to be put forwards.
- line 269 add 'levels'
- Line 277- this is helpful, but the quantity of improvement in which parameter and if placebo controlled would be helpful to add. Please critically appraise the studies included.
- Likewise for the study in next paragraph.
Rebuttal
- The comment is correct. A table (Table 1) reporting the more significant studies have been done and cited at the end of the manuscript, summarizing what cited and discussed in the present manuscript
- Fig. 3 has been modified to be more readable
- The comment might be correct but the way ALC is acting on the hypothalamus-pituitary function is complex. In any case it has been described for FHA along the paragraph. Moreover, the text refers to the studies on experimental animals that permitted to disclose where the blockade takes place. The parameters that were improved by carnitines administration, i.e. 17OHP, LH and LH response to Naloxone text, have been added and described.
- The comment is correct but it is not possible to summarize the treating doses since the whole amount of studies focused on a great amount of diseases (i.e. Alzheimer, male and female infertility, PCOS, stress induced amenorrhea, ageing, heart diseases, etc.) have used from low to high doses of carnitines or mix of ALC and L-carnitine. This review aims to give insights so that to drive clinicians to the knowledge of the role of carnitines. A more deep treating would require more pages and more efforts, not possible here.
- The comment is correct. English has been checked carefully.
- Additional references have been added in regards to ROS and more details have been added in regards to ROS production and effects of this oxidative substance.
- Capitalization has been omitted, it was a type mistake.
- Being FHA dependant from stress, high PRL plasma levels reflects the stress-induced activation. Being these studies conducted in humans, no cannulation is permitted, obviously. An hystorical amount of study have been conducted on cannulated experimental animals demonstrating that stress induces elevation not only of PRL but also of CRF, ACTH at the hypothalamic-ipophiseal level. Low estradiol plasma levels are not responsible of low PRL plasma levels in FHA. It has been demonstrated that low PRL levels in FHA occur after persistent chronic stress action combined with severe eating disorders/weight reduction. These PRL issues are not subjects of discussion in the context of this review on carnitines.
- Yes, the blunted response of cortisol to CRH test is indicative of high CRH tone at the hypothalamic level. This has been properly added in this paragraph.
- A specific description of naloxone test has been added, as requested.
- The mechanism of action of ALC has been improved, trying to condensate the effects of the molecule on the neuro-endocrine centers.
- The question is correct. All the clinical studies on carnitines effects on female reproductive function demonstrated the recovery of the menstrual/ovarian function. Safety studies were conducted and done in the 90ies when this compound was therapeutically introduced as treating option, but this is not a topic to be treated here.
- The comment is correct, however no receptors and/or precise pathway has been identified. An other specific reference was added in regards of this topic.
- The comment is correct, however no reliable data on POMC has been produced on humans neither in baseline nor under ALC treatment. The putative effects on POMC are due to the reduction of stress-induced CRF hyperactivation, as discussed in the paragraph.
- As the reviewer knows, the % of incidence of PCOS depends not only from geographic location but also from etnich characters. A specific reference was added.
- The comment is correct. This issue was expanded and the whole paragraph has been implemented with various additons in regards to the Samimi and also to Molfino studies.
- Similar changes in the paragraph where the study by Genazzani et al. was mentioned. We also added a number of new references to better disclose the details of the comments (# 16 and # 17)
Round 2
Reviewer 2 Report
Table 1 does not seem to be attached
The reference for PCOS prevalence concludes:
"The proportions of PCOS prevalence (95% CI) according to the diagnostic criteria of NIH, Rotterdam and AE-PCOS Society were 6% (5-8%, n = 18 trials), 10% (8-13%, n = 15 trials) and 10% (7-13%, n = 10 trials), respectively. "